# Quality Indicators Compliance and Survival Outcomes in Breast Cancer according to Age in a Certified Center

**DOI:** 10.3390/cancers15051446

**Published:** 2023-02-24

**Authors:** Fernando Osório, António S. Barros, Bárbara Peleteiro, Isabel Amendoeira, José Luís Fougo

**Affiliations:** 1Breast Center, São João University Hospital, 4200-319 Porto, Portugal; 2CINTESIS@RISE, GeriMHealth, Faculty of Medicine, University of Porto, 4200-450 Porto, Portugal; 3UnIC@RISE, Department of Surgery and Physiology, Cardiovascular Research and Development Unit, Faculty of Medicine, University of Porto, 4200-450 Porto, Portugal; 4Hospital Epidemiology Center, São João University Hospital, 4200-319 Porto, Portugal; 5EPIUnit, Institute of Public Health, University of Porto, 4050-600 Porto, Portugal; 6Department of Public Health and Forensic Sciences, and Medical Education, Faculty of Medicine, University of Porto, 4200-450 Porto, Portugal; 7Department of Pathology, São João University Hospital, 4200-319 Porto, Portugal; 8Ipatimup—Institute of Molecular Pathology and Immunology, University of Porto, 4200-135 Porto, Portugal

**Keywords:** breast neoplasms, neoplasms staging, age groups, aged, quality indicators, health care, combined modality therapy, outcome assessment, health care, prognosis, undertreatment

## Abstract

**Simple Summary:**

In this study, we examined how age impacts the outcomes of breast cancer by comparing three age groups: patients 45 years old or younger, patients between 46 and 69 years old, and patients 70 years old or older. Despite similar cancer staging and tumor characteristics among the age groups, the study found that older patients were prone to suboptimal treatment. Older patients also had a lower overall survival rate, but this was not related to cancer itself. Instead, we found that undertreatment was a factor that negatively impacted survival for older women with breast cancer. This study suggests that tumor characteristics and treatment compliance are more important predictors of survival than chronological age.

**Abstract:**

Age as a breast cancer (BC) prognostic factor remains debatable. Several studies have investigated clinicopathological features at different ages, but few make an age group direct comparison. The European Society of Breast Cancer Specialists quality indicators (EUSOMA-QIs) allow a standardized quality assurance of BC diagnosis, treatment, and follow-up. Our objective was to compare clinicopathological features, compliance to EUSOMA-QIs and BC outcomes in three age groups (≤45 years, 46–69 years, and ≥70 years). Data from 1580 patients with staged 0–IV BC from 2015 to 2019 were analyzed. The minimum standard and desirable target on 19 mandatory and 7 recommended QIs were studied. The 5-year relapse rate, overall survival (OS), and BC-specific survival (BCSS) were also evaluated. No meaningful differences in TNM staging and molecular subtyping classification between age groups were found. On the contrary, disparities in QIs compliance were observed: 73.1% in ≤45 years and 46–69 years women vs. 54% in older patients. No differences in loco-regional or distant progression were observed between age groups. Nevertheless, lower OS was found in older patients due to concurrent non-oncological causes. After survival curves adjustment, we underscored evidence of undertreatment impacting BCSS in ≥70 years women. Despite a unique exception—more invasive G3 tumors in younger patients—no age-specific differences in BC biology impacting outcome were found. Although increased noncompliance in older women, no outcome correlation was observed with QIs noncompliance in any age group. Clinicopathological features and differences in multimodal treatment (not the chronological age) are predictors of lower BCSS.

## 1. Introduction

Age as a prognostic factor in breast cancer (BC) remains debatable due to the lack of standardized comparative studies on the impact of multimodal treatment on age groups [1,2,3,4,5,6]. A relative consensus exists on the better prognosis of BC due to more favorable and indolent tumor biology in older women [4,5,6]. Consequently, the trend toward de-escalating therapy proceeds, often without a routine comprehensive geriatric evaluation to guide personalized multidisciplinary team (MDT) decision-making [7,8,9,10,11,12]. 

Irrespective of age, a guiding principle of BC treatment is to consider patients’ BC cure probability to minimize a preventable cause of death [13]. Survival is the key-measure of cancer treatment effectiveness [2,3,13]. The EUROCARE-5 BC sub-analysis showed significant differences in cancer survival between 29 European countries and worse outcomes in older patients [2,3]. Cancer registries should provide an estimate of population-based cancer incidence, and survival [1,2]. Ideally, a central anonymized online European database would enable standardized comparability of cancer staging, treatments, and outcomes.

The European Society of Breast Cancer Specialists (EUSOMA) certification process includes a set of quality indicators (QIs) that certified breast units should follow, enabling a standardized audit of BC diagnosis, treatment, and follow-up [14,15,16,17,18]. QIs are useful strategic tools that enhance the quality assurance of clinical practice [14,15,16,17,18]. However, they do not currently include verifiable outcome measurements to draw inferences on survival or age as a prognostic factor. The latest EUSOMA position paper update emphasizes the likely correlation between undertreatment and worse BC outcome in elderly patients. Non-penalizing compliance and flexible QIs structure to allow a personalized therapeutic approach in older adults are discussed but not formally assumed [15,16].

The current aging population and the socioeconomic impact of the associated higher cancer incidence strengthens the ethical commitment and clinical challenge of treating older women with BC through well-supported evidence [19]. Despite heterogeneous physiology and competing causes of mortality, recent studies have shown the negative impact of omitting multimodal BC treatment in older adults [7,8,9,10,20,21,22,23].

This single-center study was based on the patient cohort at the Breast Center of S. João University Hospital (BC-CHUSJ), certified by EUSOMA since 2017. The aims were to compare clinicopathological features, compliance to EUSOMA QIs and related survival outcomes in three pre-specified age groups (≤45 years, 46–69 years, and ≥70 years) from 2015 to 2019.

## 2. Materials and Methods

### 2.1. Study Design

A benchmarking audit with international standards is an ethical imperative to strive for good clinical care [14,15,16,17,18,20]. Our voluntary candidacy for EUSOMA certification, obtained in 2017, provides access to useful mandatory yearly monitoring of our daily work. The EUSOMA central database (eusomaDB) enabled quantifying QIs compliance, performing data analysis and benchmarking, and creating an opportunity for clinical research.

This observational study sought to contribute to answering the age-specific issue raised in a previous study whether the poorer outcome in older patients results from a minimalist therapeutic attitude, noted by lower QIs compliance, or whether it results from tumor (and patient) characteristics that are difficult to compare [16]. We analyzed the clinicopathological features and compliance of QIs in three different age groups over a 5-year period from 2015 to 2019.

### 2.2. Study Population

All patients with a newly diagnosed in-situ or invasive BC for 5 years (2015–2019) were included. Age groups were categorized as ≤45 years, 46–69 years, and ≥70 years according to the epidemiological profile of the population-based screening in Portugal. Patients with recurrent disease or patients who had only partial treatment at our center were excluded.

### 2.3. Data Collection

The primary outcome was to evaluate the impact of clinicopathological features and compliance to EUSOMA QIs on survival by age group. We retrospectively analyzed anonymized data from our patients in the eusomaDB. Hospital ethics committee approval to prospectively collect patients’ data in the eusomaDB was obtained (CES 93-16). The clinicopathological and immunohistochemistry (IHC) features were recorded according to EUSOMA standards. No gene expression studies for therapeutic decisions were considered.

The QIs were selected from the latest 2017 update of the EUSOMA working group [15]. From the 17 QIs main groups, we selected all the 19 mandatory and 7 recommended indicators: 9 on diagnosis, 12 on surgery and loco-regional treatment, and 5 QIs systemic-treatment-related. No QIs on staging, counselling, follow-up, and rehabilitation were considered. The QIs minimum standard and desirable target completeness were recorded in the three age groups. The completeness of registry data has been certified annually by our data-manager and validated centrally by eusomaDB.

As QIs for outcome measurements are lacking, we also analyzed the relapse rate, overall survival (OS), and BC-specific survival (BCSS). Patients were followed up until 31 December 2020. The median follow-up was 2.57 years (95% CI [1.30; 4.01]).

### 2.4. Statistical Analysis

QIs were computed for each age group using a proportional test to assess the QI performance. For continuous variables, normality was assessed by visual inspection of the data distribution, supported by QQ-plot analysis. If normally distributed, the results were summarized by the mean and standard deviation; otherwise, the median and interquartile range were described. Absolute (number) and relative (%) frequencies were reported for categorical variables. The Pearson χ2 test and Fisher’s exact tests were used to assess the differences between the independent categorical variables across the pre-defined groups. The Kruskal–Wallis rank-sum test was used to evaluate statistically significant differences between the medians of the three age groups. Survival analyses were performed using Cox proportional hazards models and the Kaplan–Meier method. A competitive risk analysis was performed to untangle BC from all-cause mortality. Internal validation was performed using bootstrap to assess the Cox models’ robustness regarding the events (death from all causes and BC). In brief, the models were run 1000 times with replacement (with event stratification), and the concordance index was gauged. All statistical analyses were performed using the *survminer* (version 0.4.9) and *survival* (version 3.2.13) packages, using R language (v.4.1.2) [24]. *p*-values < 0.05 were considered statistically significant.

## 3. Results

Patient characteristics are described in Table 1. A total of 1580 patients staged 0-IV were analyzed. A normal age distribution was observed [Appendix A], with the age extremes being proportional (19.6% were ≤45 years and 21.8% were ≥70 years). One-third of ≤45 years and 46–69 years women were overweight or obese, contrasting with 21.5% in ≥70 years women. The proportion of 46–69 years women referred from population-based screening was 30.6%. Slightly lower physical examination accuracy was observed for the detection of malignancy in this age group. More multicentric/multifocal lesions were identified at younger ages, probably due to the more widespread use of magnetic resonance imaging (MRI). The median tumor size was 17 mm in 46–69 years women, compared to 20 mm in the other two age groups. Axillary staging was statistically different: positive axilla was more prevalent in age extremes (29.0% in ≤45 years and 21.0% in ≥70 years).

No meaningful difference was observed in the proportions of 0–III TNM stages between age groups. There was a lower prevalence of stage IV at diagnosis in the youngest women (1.0% in ≤45 years versus 3.6% in 46–69 years and 3.5% in ≥70 years). Invasive grade 3 (G3) tumors were significantly more in ≤45 years women (53.3%) than in 46–69 years (35.1%) or ≥70 years (26.0%) women. The proliferation activity index (Ki67) was selectively studied in invasive G2 tumors; however, it was not performed in most cases, namely in 86.0% of older women. The proportion of luminal tumors was identical, apart from a lower prevalence of luminal-B tumors in ≥70 years women (15.0% in ≤45 years versus 8.0% in ≥70 years), as well as the triple negative (TN) tumors distribution. Concerning HER2 overexpression, we found a significant difference between age groups: 21.0% in ≤45 years, 16.0% in 46–69 years, and 10.0% (and 5.8% not assessed) in ≥70 years women.

Surgery, with a predominance of breast conservative surgery, was the first treatment option in all age groups. Most significant was the difference in the omission of surgery: 17.4% in older women, 3.0% in 46–69 years, and 0.6% in ≤45 years. The proportion of younger women who began their treatment with neoadjuvant chemotherapy (37.4%) was identical to that of older women who started with endocrine therapy (32.3%). Adjuvant chemotherapy or radiotherapy was significantly less undertaken in older patients (12.8% and 47.7%, respectively).

To emphasize the Table 1 contingency cross-table significance values that are driving the observed significances, we provide in Appendix A, the adjusted standardized residuals.

The QIs compliance is given in Table 2. Of the 19 selected mandatory QIs, we reached threshold requirements in 9 QIs in all three age groups, in 3 of them with 100% completeness (in cancers with a pre-operative diagnosis, in cancers discussed by a multidisciplinary team, and in HER2 invasive cancers treated with neo-adjuvant chemotherapy plus trastuzumab). In contrast, we did not achieve the minimum requirement in mandatory 4 QIs in any age group (in pathological and IHC characterization of invasive and non-invasive cancers and pN+ invasive cancers receiving post-mastectomy radiotherapy). Regarding the recommended QIs, we achieved 100% completeness in invasive cancers’ clinical and imagiological axillary staging in all age groups. Still, we failed to meet the minimum standard in 2 QIs, in the waiting time until first treatment and in immediate reconstruction after mastectomy.

The QIs compliance was 73.1% in the youngest and 46–69 years women, and 54.0% in the older patients. In the ≤45 years group, we attained the desired target in 11 of 19 mandatory QIs and 2 of 7 recommended QIs. The minimum standard was reached additionally in 3 mandatory QIs and 3 recommended QIs. In the 46–69 years group, the desirable target was achieved in 13 mandatory QIs and in 2 recommended QIs and the minimum standard was reached in 4 QIs (1 mandatory and 3 recommended QIs). In the ≥70 years group, we reached the desired target in 8 obligatory QIs and 2 recommended QIs and the minimum standard in another 3 mandatory QIs and 1 recommended QI.

The 5-year loco-regional relapse showed no disparities between age groups. The same was found for distant progression with no significant difference in the distribution of sites of metastatic disease between younger or older patients [Appendix A]. We observed a considerable difference in ≥70 years women’s 5-year OS (Figure 1). The higher mortality found in older patients was mainly explained by concurrent non-oncological causes of death [Appendix A] and faded when considering the BCSS (Figure 1). Survival differences according to age are mainly explained by differences in tumor grade (but not tumor size), axillary stage and combined multimodal treatment [Figure 2].

## 4. Discussion

Chronological age should not be a determinant of BC prognosis or treatment [7,25]. This is well-defined in the youngest women thanks to several consensus meetings, but despite similar experts’ efforts, a therapeutic stigma on advanced age persists in daily practice nowadays [7,25,26,27,28,29,30]. The postulated consensus that advanced age is associated with a more favorable tumor biology (more luminal tumors, fewer TN or HER2 tumors, lower proliferative rates) that allows less intensive treatment is contradicted by the observed poorer outcome of BC in older patients [8,26,31,32,33,34,35,36].

Although several studies have investigated clinicopathological features at different ages, few made an age group direct comparison [2,4,5,6,16,21,22,23,31,32,33,34,35,36]. Our cohort showed no statistically significant age pattern disparities when compared to other studies [1,3,4,5,6,31,32,33,34,35,36]. As reported in previous studies, the median tumor size corresponded to a cT1 stage in all age groups [1,36]. More relevant was the significant difference regarding multicentric/multifocal lesions more frequently found at younger ages. Contrary to several studies [1,31,32,33,34,35], on TNM pathologic staging and molecular subtypes, we found no significant differences between age groups. There were three exceptions in younger patients: a more prevalent positive axilla, a lower prevalence of stage IV at diagnosis, and a higher prevalence of luminal-B tumors. However, a more aggressive biology of BC in younger patients was only significantly observed in one pathological feature: a higher prevalence of invasive G3 tumors. Likewise [1,5,6,31,32,33,34,35], no significant differences were found between age groups regarding hormonal status. The same was observed concerning the proportion of TN tumors. As for HER2-positive tumors, a difference was seemingly uncovered, though inconclusive, since HER2 over-expression was not studied in 5.8% of older patients. Like Ki67 (not performed in 85.9% of ≥70 years), this incomplete IHC study regarding HER2 status in older adults discloses a potential preconceived advanced age intention-not-to-treat [26]. Considering genetic assessment, there was inevitably an age bias, as more younger patients were studied, and more genetic mutations were identified in ≤45 years women. Even so, two BRCA mutations were identified in ≥70 years patients.

International guidelines, such as the EUSOMA QIs, should be regarded as recommendations for excellence in clinical care [7,14,15,16,17,18]. The obtained EUSOMA certification with annual monitoring of their QIs helped to improve our quality control. The QIs also validated our cohort conclusions about diagnostic and therapeutic options. Some disparities in the QIs compliance were observed between age groups. We corroborate the reported difficulty of covering every case on QIs [17,18] since, over time, some patients needed a non-standard approach which was not easy to audit. 

As in the previously cited study devoted to the variation in compliance to EUSOMA QIs by age [16], we noted a significant lower compliance to QIs in older patients. This did not occur in the other two age groups. Concerning the BC diagnosis (initial clinical and imagiological study, axillary staging, and pre-operative needle-biopsy confirmation of cancer), we reached the mandatory QIs target recommendations (with a minor exception in younger patients). We did not match the QIs minimum requirements in any age group in the characterization of in situ or invasive disease. Still, the pathological requirements in those two QIs substantially changed during our study period. From the recommended diagnostic QIs, we achieved the minimum requirements for the MRI study and the genetic counseling referral apart from older women. The most astounding of our audit, which demanded corrective action, was the noncompliance in any age group of the non-obligatory ≤6 weeks’ waiting time to start treatment. We reached almost complete QIs compliance regarding surgical treatment in all age groups with two exceptions: the omission of sentinel node biopsy in nine older women with cN0 invasive BC and not having reached the immediate reconstruction after mastectomy QIs target, most evidently in older women. Regarding radiation therapy, noncompliance in three mandatory QIs was observed in ≥70 years women. The minimum requirements for post-mastectomy radiotherapy were not achieved in all age groups. Endocrine therapy was appropriate in all age groups, but existing QIs were not discriminative for chemotherapy in luminal tumors. Noncompliance of mandatory QIs regarding systemic treatment of TN tumors was observed in older patients. Otherwise, in the HER2-positive tumors we reached the two QIs target recommendations for all age groups. In summary, our MDT discussion, surgical treatment (except for immediate reconstruction), endocrine therapy, and anti-HER2 therapy were above the mandatory thresholds in all age groups. However, this was not observed for adjuvant radiotherapy in pN+ disease or adjuvant chemotherapy in TN tumors, mainly for ≥70 years women [37]. Nonetheless, a possible correlation between EUSOMA QIs noncompliance and possible undertreatment in older patients could not be concluded, because neither the former is sufficiently discriminatory in monitoring BC best practices and outcomes [15,16,18], or includes age-specific standards, such as geriatric covariates, nor is the latter consensually defined in the literature [25,26]. The EUSOMA QIs assess compliance of mandatory variables to adequate BC diagnosis and treatment but are not helpful as a tool for predicting an objective outcome, such as the 5-year BCSS.

Without irrefutable evidence [4,5,6,20,31,32,33,34,35], and despite the observed differences in multimodal treatment, the clinical outcome data reported in our cohort showed no significant difference in 5-year loco-regional relapse or distant progression between the three age groups. Nevertheless, a significant difference was found regarding vital status, and concurrent non-oncological causes of death could not be the only explanation for the higher mortality in older patients [23]. After adjusting survival curves for the Cox hazard proportional model, we underscored evidence of undertreatment impacting BCSS in ≥70 years women [Figure 2 and Appendix A]. As previously reported [8,9,10,11,12,21,22,37,38,39,40], the older age subgroup was being comparatively less treated. A higher proportion of omission of surgery and a less frequent option for adjuvant chemotherapy or radiotherapy was observed with a deleterious impact on survival [Figure 2]. Multivariable analysis showed no significant difference regarding age group (HR 0.87, *p* = 0.698), but evidence of undertreatment impacting survival: surgery (HR 0.11, *p* < 0.001) and adjuvant radiotherapy (HR 0.37, *p* = 0.004) allow a reduction of BC-specific mortality risk. Furthermore, an analysis of the competing risks [Appendix A] showed that younger women die more from cancer. In older women, there is no difference between causes of death in the first two years. Nonetheless, they die more from non-oncological causes after that period, reversing the trend observed in younger women. This underscores the significance of scrutinizing BCSS in addition to OS, and considering, through appropriate multimodal treatment, BC as a preventable cause of mortality in older patients.

Finally, Cox model robustness was assessed using bootstrapping with replacement. One thousand runs were performed for each event (death from all causes and BC), and the concordance index (C-index over 0.8) suggests robust models for both events [Appendix A].

## 5. Conclusions

Our study sought to contribute to demystifying the common misconception of age-specific differences in BC biology impacting outcomes, which we did not observe in daily practice. Despite a unique exception—a higher prevalence of invasive G3 tumors in ≤45 years patients—a more aggressive biology was not observed at younger ages nor the contrary in older women. Although there was increased noncompliance in older women, no outcome correlation was observed with QIs noncompliance in any age group.

Chronological age should not be considered a prognostic factor in BC, as clinicopathological features and multimodal treatment differences are the main predictors of lower BCSS in older patients. For this age subgroup, earlier diagnosis, and personalized treatment—supported not by a frailty demand (to de-escalate therapy) but by a multidimensional geriatric assessment to pursue biological age and socio-family circumstance—are demanded.

## Figures and Tables

**Figure 1 cancers-15-01446-f001:**
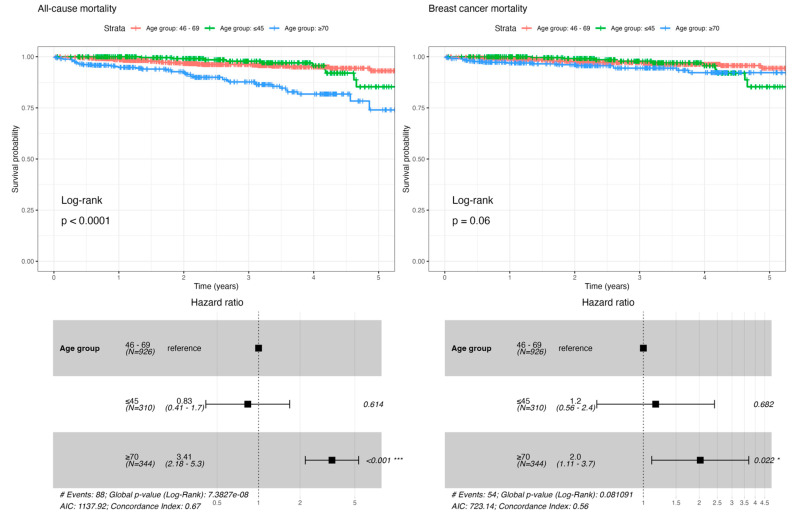
All-cause mortality and BC-specific mortality by age group (* *p* < 0.05, *** *p* < 0.001).

**Figure 2 cancers-15-01446-f002:**
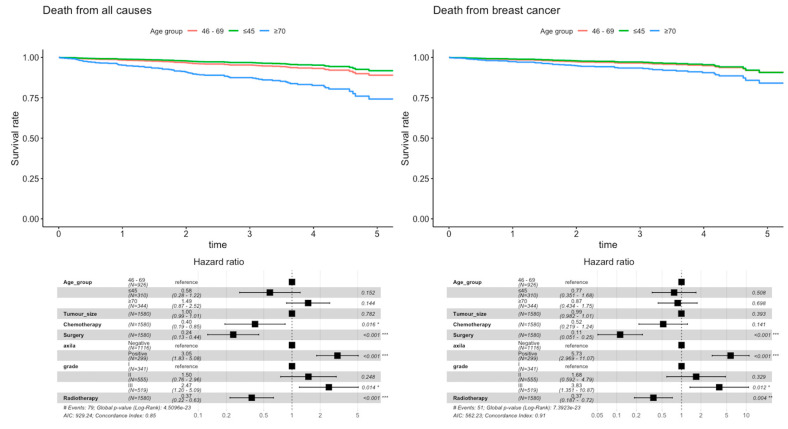
Adjusted Cox proportional hazard regression model for all-cause mortality and BC-specific mortality (* *p* < 0.05, ** *p* < 0.01, *** *p* < 0.001).

**Table 1 cancers-15-01446-t001:** Cohort clinicopathological characteristics stratified by age group.

Characteristic	N	≤45 y, N = 310 ^1^	46–69 y, N = 926 ^1^	≥70 y, N = 344 ^1^	*p*-Value ^2^
Sex	1580				0.6
Woman		310 (100%)	923 (99.7%)	344 (100%)	
Man		0 (0%)	3 (0.3%)	0 (0%)	
Body Mass Index (BMI)	1580				<0.001
Not evaluated		91 (29.4%)	421 (45.5%)	235 (68.3%)	
<18.5–24.9		133 (42.9%)	186 (20.0%)	35 (10.2%)	
25.0–29.9		59 (19.0%)	199 (21.5%)	39 (11.3%)	
≥30.0		27 (8.7%)	120 (13.0%)	35 (10.2%)	
Referral from screening programme	1580	3 (1.0%)	283 (30.6%)	15 (4.4%)	<0.001
Clinical examination/Suspicious of malignancy (yes)	1580	258 (83.2%)	614 (66.3%)	274 (79.7%)	<0.001
Side location of the lesion	1580				0.5
Left		158 (51.0%)	497 (53.7%)	173 (50.3%)	
Right		152 (49.0%)	429 (46.3%)	171 (49.7%)	
Disease extent by imaging or clinical examination	1580				<0.001
Localized		259 (83.5%)	837 (90.4%)	324 (94.2%)	
Multicentric/Multifocal		51 (16.5%)	89 (9.6%)	20 (5.8%)	
Magnetic Resonance Imaging (RMI) (yes)	1580	225 (72.6%)	375 (40.5%)	51 (14.8%)	<0.001
Median Tumor size by imaging or physical examination [mm, (IQR)]	1551	20 (15, 32)	17 (11, 27)	20 (12, 30)	<0.001
Axillary staging (including only invasive tumors)	1415				0.001
Negative		204 (71.1%)	664 (81.0%)	248 (79.2%)	
Positive		83 (28.9%)	151 (19.0%)	65 (20.8%)	
TNM stage	1580				<0.001
0		22 (7.1%)	107 (11.6%)	28 (8.1%)	
I		125 (40.3%)	483 (52.1%)	154 (44.8%)	
II		143 (46.1%)	288 (31.1%)	136 (39.5%)	
III		17 (5.5%)	15 (1.6%)	14 (4.1%)	
IV		3 (1.0%)	33 (3.6%)	12 (3.5%)	
Invasive histological type at biopsy (including invasive and microinvasive tumors)	1423				0.2
Ductal/No Special Type (NST)		240 (83.3%)	658 (80.3%)	239 (75.6%)	
Lobular		23 (8.0%)	82 (10.0%)	35 (11.1%)	
Other		25 (8.7%)	79 (9.7%)	42 (13.3%)	
Final pathology	1580				<0.001
In situ		22 (7.1%)	107 (11.5%)	28 (8.1%)	
Invasive (including microinvasive)		236 (76.1%)	762 (82.3%)	314 (91.3%)	
Invasive at biopsy only with pathological complete response		52 (16.8%)	57 (6.2%)	2 (0.6%)	
Modified Bloom–Richardson Grade (including only invasive tumors)	1415				<0.001
G1		41 (14.3%)	209 (25.6%)	91 (29.0%)	
G2		93 (32.4%)	320 (39.3%)	142 (45.4%)	
G3		153 (53.3%)	286 (35.1%)	80 (25.6%)	
Lymphovascular invasion (including operated invasive tumors)	1283				<0.001
No		154 (65.8%)	610 (81.0%)	250 (84.5%)	
Yes		80 (34.2%)	143 (19.0%)	46 (15.5%)	
Oestrogen receptor status (including only invasive tumors)	1415				0.044
Not performed		1 (0.3%)	0 (0%)	0 (0%)	
Negative		63 (22.0%)	135 (16.6%)	48 (15.3%)	
Positive		223 (77.7%)	680 (83.4%)	265 (84.7%)	
Progesterone receptor status (including only invasive tumors)	1415				0.4
Not performed		0 (0%)	6 (0.7%)	4 (1.3%)	
Negative		89 (31.0%)	258 (31.7%)	92 (29.4%)	
Positive		198 (69.0%)	551 (67.6%)	217 (69.3%)	
HER2 overexpression (including only invasive tumors)	1415				<0.001
Not Performed		0 (0%)	0 (0%)	18 (5.8%)	
Negative		226 (78.7%)	685 (84.0%)	263 (84.0%)	
Positive		61 (21.3%)	130 (16.0%)	32 (10.2%)	
Proliferation activity index (Ki67) in invasive G2 tumors	555				<0.001
Not performed		57 (61.3%)	203 (63.4%)	122 (85.9%)	
<5%		4 (4.3%)	21 (6.6%)	4 (2.8%)	
5–30%		23 (24.7%)	82 (25.6%)	13 (9.2%)	
>30%		9 (9.7%)	14 (4.4%)	3 (2.1%)	
Molecular subtyping (including only invasive tumours)	1415				<0.001
Luminal A-like		182 (63.4%)	585 (71.8%)	224 (71.5%)	
Luminal B-like		43 (15.0%)	94 (11.5%)	26 (8.3%)	
HER2 missing		0 (0%)	0 (0%)	18 (5.8%)	
HER2 positive		20 (7.0%)	43 (5.3%)	11 (3.5%)	
Triple Negative		42 (14.6%)	93 (11.4%)	34 (10.9%)	
BRCA1 + BRCA2	1580				<0.001
No genetic assessment		123 (39.7%)	784 (84.7%)	334 (97.1%)	
Negative		173 (55.8%)	129 (13.9%)	8 (2.3%)	
Positive		14 (4.5%)	13 (1.4%)	2 (0.6%)	
First treatment	1580				<0.001
Surgery		190 (61.3%)	717 (77.4%)	210 (61.0%)	
Neoadjuvant chemotherapy		116 (37.4%)	169 (18.3%)	16 (4.7%)	
Primary endocrine therapy		4 (1.3%)	37 (4.0%)	111 (32.3%)	
Support treatment/Surveillance/ Patient refusal		0 (0%)	3 (0.3%)	7 (2.0%)	
Surgery	1580				<0.001
Breast conservative surgery		170 (54.9%)	655 (70.8%)	194 (56.4%)	
Mastectomy		138 (44.5%)	243 (26.2%)	90 (26.2%)	
No surgery		2 (0.6%)	28 (3.0%)	60 (17.4%)	
Endocrine therapy (yes)	1580	237 (76.5%)	721 (77.9%)	271 (78.8%)	0.8
Chemotherapy (yes)	1415	96 (33.4 %)	260 (31.9%)	40 (12.8%)	<0.001
Radiotherapy (yes)	1580	211 (68.1%)	687 (74.2%)	164 (47.7%)	<0.001

^1^ N (%); Median (IQR). ^2^ Fisher’s exact test for count data with simulated *p*-value (based on 2000 replicates); Pearson’s chi-squared test; Kruskal–Wallis rank sum test.

**Table 2 cancers-15-01446-t002:** EUSOMA Quality Indicators compliance by age group.

						≤45 Years	46–69 Years	≥70 Years
Evidence	Mand/Recom	Min. Req. (%)	Target (%)	Cases/Total	Result (%)	95% I.C.	Cases/Total	Result (%)	95% I.C.	Cases/Total	Result (%)	95% I.C.
1	Cancers who underwent pre-operative physical examination, mammography/ultrasound of both breasts and axillae	III	Mandatory	**>90**	**>95**	289/310	**93.2**	(89.7; 95.7)	895/926	**96.7**	(95.2; 97.7)	328/344	**95.3**	(92.4; 97.2)
3a	Invasive cancers who underwent axillary staging by ultrasound +/− FNA/CNB	III	Recommended	**85**	**95**	287/287	**100**	(98.4; 100)	815/815	**100**	(99.4; 100)	313/313	**100**	(98.5; 100)
3b	Cancers (invasive or in situ) with a pre-operative confirmed diagnosis (B5 or C5)	III	Mandatory	**85**	**90**	310/310	**100**	(98.4; 100)	926/926	**100**	(99.5; 100)	344/344	**100**	(98.6; 100)
4a	Invasive cancers with histological type, grading, ER/HER2, pN, margins, vascular invasion & size recorded	II	Mandatory	**>95**	**>98**	243/288	**84.4**	(79.5; 88.3)	748/819	**91.3**	(89.1; 93.1)	245/316	**77.5**	(72.4; 81.9)
4b	Non-invasive cancers with histological pattern, grading, size, margins & ER recorded	II	Mandatory	**>95**	**>98**	17/22	**77.3**	(54.2; 91.3)	65/107	**60.7**	(50.8; 69.9)	12/28	**42.9**	(25.0; 62.6)
5	Waiting time ≤ 6 weeks between the date of first diagnostic examination (mammogram/ultrasound) and surgery/other treatment	IV	Recommended	**80**	**90**	194/310	**62.6**	(56.9; 67.9)	476/926	**51.4**	(48.1; 54.7)	202/344	**58.7**	(51.3; 62.0)
6a	Cancers examined preoperatively by MRI (excluding PST)	IV	Recommended	**10**	**NA**	102/164	**62.2**	(54.3; 69.5)	193/638	**30.3**	(26.7; 34.0)	38/294	**12.9**	(9.4; 17.4)
6b	Cancers treated with PST undergoing MRI	IV	Recommended	**60**	**90**	107/123	**87.0**	(79.4; 92.2)	155/177	**87.6**	(81.6; 91.9)	10/19	**52.6**	(29.5; 74.8)
7	Cancers refered for genetic counselling	IV	Recommended	**10**	**NA**	187/310	**60.3**	(54.6; 65.8)	142/926	**15.3**	(13.1; 17.9)	10/344	**2.9**	(1.5; 5.5)
8	Cancers discussed pre and postoperatively by a MDT	III	Mandatory	**90**	**99**	310/310	**100**	(98.5; 100)	926/926	**100**	(99.5; 100)	344/344	**100**	(98.6; 100)
9a	Invasive cancers receiving just 1 operation (excl. reconstruction)	II	Mandatory	**80**	**90**	276/288	**95.8**	(92.6; 97.7)	784/819	**95.7**	(94.0; 97.0)	307/316	**97.2**	(94.5; 98.6)
9b	DCIS receiving just 1 operation (excl. reconstruction)	II	Mandatory	**70**	**90**	20/22	**90.9**	(69.4; 98.4)	87/99	**87.9**	(79.4; 93.3)	25/28	**89.3**	(70.6; 97.2)
9c	Immediate reconstruction after mastectomy	III	Recommended	**40**	**40**	54/138	**39.1**	(31.0; 47.8)	79/243	**32.5**	(26.7; 38.8)	4/90	**4.4**	(1.4; 11.6)
10a	M0 invasive cancers receiving postoperative RT after BCT	I	Mandatory	**90**	**95**	145/154	**94.2**	(88.9; 97.1)	542/565	**95.9**	(93.9; 97.3)	129/160	**80.6**	(73.5; 86.4)
10b	Cancers ≥ pN2a+ receiving post-mastectomy RT	I	Mandatory	**90**	**95**	6/7	**85.7**	(42.0; 99.2)	11/18	**61.1**	(36.1; 81.7)	1/4	**25.0**	(1.3; 78.1)
10c	Cancers pN1 receiving post-mastectomy RT	I	Mandatory	**70**	**85**	18/35	**51.4**	(34.3; 68.3)	37/60	**61.7**	(48.2; 73.6)	6/19	**31.6**	(13.6; 56.5)
11a	Invasive cancers cN0 who underwent SLNB only (excluding PST)	I	Mandatory	**90**	**95**	146/147	**99.3**	(95.7; 100)	560/563	**99.5**	(98.3; 99.9)	73/82	**89.0**	(79.7; 94.5)
11b	No more than 5 nodes excised in invasive cancers who underwent SLNB	I	Recommended	**90**	**95**	184/188	**97.9**	(94.3; 99.3)	573/590	**97.1**	(95.3; 98.3)	170/175	**97.1**	(93.1; 98.9)
11c	Invasive cancers ≤ 3cm (incl. DCIS component) who underwent BCT (BRCA patients excluded)	I	Mandatory	**70**	**85**	80/105	**76.2**	(66.7; 83.7)	412/473	**87.1**	(83.7; 89.9)	119/146	**81.5**	(74.1; 87.3)
11d	Non-invasive cancers ≤ 2cm treated with BCT	II	Mandatory	**80**	**90**	3/6	**50.0**	(18.8; 81.2)	46/48	**95.8**	(84.6; 99.3)	14/17	**82.4**	(55.8; 95.3)
11e	DCIS who do not undergo axillary clearance	II	Mandatory	**97**	**99**	20/20	**100**	(80.0; 100)	102/103	**99.0**	(93.9; 99.9)	23/24	**95.8**	(76.9; 99.8)
12	Endocrine sensitive invasive cancers receiving HT	I	Mandatory	**85**	**90**	217/223	**97.3**	(94.0; 98.9)	662/680	**97.4**	(95.8; 98.4)	259/263	**98.5**	(95.9; 99.5)
13a	Invasive cancers ER negative (T > 1cm or N+) receiving adjuvant CT	I	Mandatory	**85**	**95**	59/60	**98.3**	(89.9; 99.9)	117/121	**96.7**	(91.2;98.9)	27/44	**61.4**	(45.5; 75.3)
13b	Invasive cancers HER2 positive (T > 1cm or N+) treated with adjuvant CT who received adjuvant trastuzumab	I	Mandatory	**85**	**95**	9/9	**100**	(62.9; 100)	45/47	**95.7**	(84.3; 99.3)	10/10	**100**	(65.5; 100)
13c	Invasive cancers HER2 positive treated with neo-adjuvant CT who received neo-adjuvant trastuzumab	I	Mandatory	**90**	**95**	29/29	**100**	(85.4; 100)	44/44	**100**	(90.0; 100)	2/2	**100**	(19.8; 100)
13d	Inflammatory breast cancer who received neo-adjuvant CT	II	Mandatory	**90**	**95**	3/3	**100**	(31.0; 100)	6/7	**85.7**	(42.0; 99.2)	1/1	**100**	(5.5; 100)

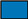
 Diagnostic Qis; 
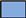
 Loco-regional treatment Qis; 
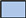
 Systemic treatment Qis; 

 Mandatory requirements Qis; 
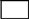
 Recommended requirements Qis; 

 Minimum standard Qis; 
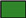
 Desirable target Qis; 
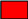
 QIs noncompliance.

## Data Availability

All research data was archived in the hospital’s database and can be provided, if requested, to the corresponding author.

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
