# Peer review of "Quality Indicators Compliance and Survival Outcomes in Breast Cancer according to Age in a Certified Center"

_cancers, 2023, doi:10.3390/cancers15051446_

Round 1

Reviewer 1 Report

In the publication, the authors use EUSOMA QI, which should serve to improve the diagnosis and therapy of breast cancer in a certified center. It is a very careful prospective process of data collection from 2015-2019 and their analysis according to 19 mandatory and 7 recommended QIs. It is not clearly explained what criteria were used to select these QIs for this topic? Would any of the unused criteria lead to a different result? It is positive that the most complete results are in the field of operative therapy, but that the other parts of diagnostics and therapy significantly affect the final evaluation and should be of the same quality. First of all, breast cancer is a systemic disease in which biological characteristics play a decisive role in, as the authors call it, the survival outcome.  In any case, the publication summarized by the author deserves to be published after the explanation, not only because it brings specific results for the certified center, but also serves as a motivation for a more complete documentation of all parts of breast cancer treatment.

Author Response

We thank the reviewer for the opportunity to revise our work.

We believe with the reviewer's suggestions the manuscript has improved greatly.

In the publication, the authors use EUSOMA QI, which should serve to improve the diagnosis and therapy of breast cancer in a certified center. It is a very careful prospective process of data collection from 2015-2019 and their analysis according to 19 mandatory and 7 recommended QIs.

Q1: It is not clearly explained what criteria were used to select these QIs for this topic?

A1: We thank the reviewer for pointing out this issue. From all the EUSOMA QIs we chose those related to the characterization of the diagnosis and multimodal treatment of breast cancer. We have chosen all the mandatory QIs and some of the QIs recommended but not mandatory in the annual certifications we have undergone since 2017.

Q2: Would any of the unused criteria lead to a different result? 

A2: As described in the Methods section, QIs on staging, counseling, follow-up, and rehabilitation were not considered because they were outside the scope of the study. Therefore, no different results would be expected if these QIs would have been included.

Q3: It is positive that the most complete results are in the field of operative therapy, but that the other parts of diagnostics and therapy significantly affect the final evaluation and should be of the same quality. 

A3: As mentioned in the manuscript, EUSOMA QIs focus mainly on locoregional treatment criteria (surgery and radiotherapy). We hold the view that these guidelines may not adequately address certain medical treatments, specifically chemotherapy in luminal breast cancer.

The EUSOMA QIs assess compliance of mandatory variables to adequate diagnosis and treatment in breast cancer but are not helpful as a tool for predicting an objective outcome, such as the 5y-Breast Cancer Specific Survival (BCSS). 

First of all, breast cancer is a systemic disease in which biological characteristics play a decisive role in, as the authors call it, the survival outcome. In any case, the publication summarized by the author deserves to be published after the explanation, not only because it brings specific results for the certified center, but also serves as a motivation for a more complete documentation of all parts of breast cancer treatment.

The authors would like to thank the reviewer for the questions raised, the proposals for improvement, and the kind critical appraisal of our manuscript.

Reviewer 2 Report

The authors performed an elegant study on an interesting subject, given the increasing increase in life expectancy, especially in women.

As points to improve, I request:

Clarify whether the analyses, mainly contingency, related to Table 1 involved data not evaluated or whether they were excluded.

Still regarding the contingency analyses, provide a table with the adjusted standardized residuals to verify the groups with statistically significant differences.

Author Response

We thank the reviewers for the opportunity to revise our work.

We believe with the reviewer's suggestions the manuscript has improved greatly.

The authors performed an elegant study on an interesting subject, given the increasing increase in life expectancy, especially in women.

As points to improve, I request:

 Q1: Clarify whether the analyses, mainly contingency, related to Table 1 involved data not evaluated or whether they were excluded.

A1: Table 1 comprises all our cohort's descriptive variables, which are usually collected in the clinicopathological characterization of breast cancer. We have examined all the variables that were included. Furthermore, none of the variable levels has been excluded from the contingency cross-table variables displayed in Table 1.

Q2: Still regarding the contingency analyses, provide a table with the adjusted standardized residuals to verify the groups with statistically significant differences.

A2: Your suggestion is greatly appreciated, as it will enhance the clarity of the message for the reader. Consequently, we have added a supplementary table (S3 in the supplementary materials) in the revised manuscript that displays the adjusted standardized residuals.

Please see the uploaded doc. for the new S3 table.

Reviewer 3 Report

This study focuses on age-dependent compliance and survival outcomes in breast cancer patients.

Simple summary and abstract look similar. Authors should change the expression in simple summary.

Table 2 is not clear, and is very difficult to see. Authors should modify it.

Letters of Fig1 and 2 are also too small to see clearly.

Authors suggested that they underscored evidence of undertreatment impacting BCSS in women with >70 years. Which result is the evidence? Are there results showing that survival differed significantly depending on choice of the treatment in these women?

Multivariate analyses including age for survival are needed to see if age really influences the survival.

Overall, results of this study are not surprising or novel.  

Author Response

We thank the reviewer for the opportunity to revise our work.

We believe with the reviewer's suggestions the manuscript has improved greatly.

This study focuses on age-dependent compliance and survival outcomes in breast cancer patients.

Q1: Simple summary and abstract look similar. Authors should change the expression in simple summary.

A1: The authors agree with the reviewer. The summary has been revised to use simpler language that is suitable for a wider audience.

“In this study, we examined how age impacts the outcomes of breast cancer by comparing three age groups: patients 45 years old or younger, patients between 46 and 69 years old, and patients 70 years old or older. Despite similar cancer staging and tumor characteristics among the age groups, the study found that older patients were prone to suboptimal treatment. Older patients also had a lower overall survival rate, but this was not related to cancer itself. Instead, we found that undertreatment was a factor that negatively impacted survival for older women with breast cancer. This study suggests that tumor characteristics and treatment compliance are more important predictors of survival than chronological age.”

Q2: Table 2 is not clear, and is very difficult to see. Authors should modify it.

A2: We tried to show graphically all the information about the EUSOMA QIs compliance in table 2.

We have previously deposited an editable version of the Excel file as supplementary material (zenodo.org doi:10.5281/zenodo.7577252) for better readability. We also included in the manuscript a high-resolution image of the Excel table (TIFF format at 300 DPIs).

Q3: Letters of Fig1 and 2 are also too small to see clearly.

A3: To facilitate better comparability, we deemed it was crucial to present the mortality curves (both overall and cancer-specific) were shown of figures 1 and 2 in parallel. To enhance clarity, we have made improvements to the Kaplan-Meier curves by eliminating the risk table. Consequently, our revised manuscript now includes an updated and refined image.

Q4: Authors suggested that they underscored evidence of undertreatment impacting BCSS in women with >70 years. Which result is the evidence? Are there results showing that survival differed significantly depending on choice of the treatment in these women?

A4: As mentioned in the manuscript, we have found that the older age subgroup receives comparatively less treatment, adversely affecting their survival rate, as demonstrated by our results.

From figure 2 we can deduce that surgery (HR 0.11), radiotherapy (HR 0.37), and chemotherapy (HR 0.52) allows patients to reduce the risk of death. By not doing so, we create undertreatment. Furthermore, an analysis of the competing risks illustrated in figure S2 shows that younger women die more from cancer. In older women, there is no difference between causes of death in the first two years but afterwards, they die more from non-oncological causes, reversing the trend observed in younger women. We can thus conclude that if older women were adequately treated, we could minimize breast cancer as a cause of premature mortality.

Q5: Multivariate analyses including age for survival are needed to see if age really influences the survival.

A5: The multivariable analysis includes age, as shown in Figure 2. We did not observe a significant difference regarding the effect of the age group (HR p>0.05).

Q6: Overall, results of this study are not surprising or novel.

A5: The authors would like to thank the reviewer for pertinent questions and suggestions for the improvement of the manuscript.

We believe our conclusion is explicit. Our study aimed to shed light on a recurring trend in the literature that suggests a more favorable breast cancer tumor biology among elderly women, thereby contributing to its demystification. We also reinforce the negative impact on the prognosis of multimodal undertreatment in older women.

Round 2

Reviewer 3 Report

I do not have any more comments.